# Tokens on Demand: Token Condensation as Training-free Test-time Adaptation

## Abstract

In this work, we introduce **T**oken **C**ondensation as **A**daptation (**TCA**), a training-free approach designed to mitigate distribution shifts encountered by vision-language models (VLMs) during test-time inference. TCA bridges distribution gaps at the patch level by condensing image tokens that exhibit low attentiveness to the `<cls>` token. Recognizing the `<cls>` token may correspond to universal concepts, TCA identifies and tracks the most reliable `<cls>` tokens that align specifically with target classes from historical data streams. To achieve this, we propose a context token reservoir (CTR), which retains tokens with the lowest uncertainty as "anchors" to guide the preservation of class-relevant tokens during inference. These anchors, in turn, act as token-level classifiers to correct VLM predictions and improve visual-text alignment. Utilizing anchors sampled from CTR, TCA condenses tokens through two operations: (1) pruning *class-irrelevant* tokens that consistently rank low across all attention heads to reach cross-head consensus on their irrelevance, and (2) merging the remaining *class-ambiguous* tokens into representative centers using coreset selection, maintaining linear computational complexity. As the first method to explore token efficiency in test-time adaptation, TCA consistently demonstrates superior performance across cross-dataset and out-of-distribution adaptation tasks, reducing GFLOPs by 12.2% to 48.9% while achieving accuracy improvements up to 21.4% against the strongest baseline without introducing additional parameters.

## 1 Introduction

Online test-time adaptation (TTA) (Wang et al., 2023c) has emerged as a promising strategy to handle distribution shifts encountered during inference (Liang et al., 2023). TTA dynamically fine-tunes pretrained models on unlabeled data batches, enhancing generalization by aligning intermediate-layer batch statistics (Niu et al., 2023), optimizing for first-order flatness in the loss landscape (Foret et al., 2021), promoting self-supervised consistency across augmentations (Zhang et al., 2022), or tracking model historical weights (Lee & Chang, 2024). Despite the success of traditional TTA methods, they often require computationally expensive tuning of the backbone's parameters. This challenge is further amplified in vision-language models (VLMs), which consist of vast parameter sets and require large batch sizes (*e.g.,* 256) for stabilizing adaptation (Döbler et al., 2024).

Test-time prompting (TPT) offers a more efficient alternative for TTA by shifting adaptation focus to the *language* side of VLMs, learning *a small set* of task-specific context prompts for downstream tasks while freezing the visual backbone. Nevertheless, TPT largely overlooks the impact of *visual* distribution shifts. Adapting to high-variance target images through prompts often relies on external source data (Samadh et al., 2023) or extensive data augmentation (Feng et al., 2023) (*e.g.,* 60× more AugMix or diffusion-based synthetic samples). In strict online TTA settings, where the batch size is constrained to one, this reliance on augmentation significantly inflates computational costs, leading to a 60× rise in GFLOPs compared single-sample processing (*i.e.,* 1108.61 vs. 17.59 GFLOPs). The need for gradient backpropagation during inference further increases the computation burden, making exiting TPT suboptimal for many resource-constrained applications.

In this paper, we attempt to tackle visual shifts at a *patch* level by introducing a novel approach named **Token Condensation as Adaptation (TCA)**. TCA allows the model to adapt on the fly to unseen target domains while accelerating VLM inference by 20% fewer GFLOPs without requiring

Figure 1: **An illustration of the proposed Token Condensation as Adaptation (TCA).** To better adapt visual embeddings to text embeddings during test-time, TCA selectively prunes and merges tokens *(top-right)* with low attentiveness to the `<cls>` token. A historical `<cls>` token with the lowest uncertainty is sampled from the context-aware token reservoir (CTR) and serves as an "anchor" to move visual embeddings $\mathbf{z}_t$ toward text embeddings $\mathbf{t}_c$. These anchors act as token-level classifiers, refining predictions through the logit correction step.

additional training. To understand where the visual shifts came from, we conducted a *leave-one-out* preliminary study to evaluate the impact of dropping patch tokens on visual-text alignment, as depicted in Figure 2a. The results reveal that discarding tokens with lower attentiveness to its `<cls>` token may not harm, and can even improve run-time predictions. This motivates us to focus on *condensing* two key types of tokens for adaptation: (1) *class-irrelevant background tokens*, which may mislead VLMs by emphasizing non-essential regions that differ from the pre-training data, leading to erroneous predictions; (2) *class-ambiguous object tokens*, such as cat whiskers or fur, which can overlap across other classes and disperse visual embeddings.

However, relying solely on the current sample's `<cls>` token for token condensation may not be optimal, as it tends to be overly *universal* and may not specifically align with the target classes. Although the text encoder in VLMs offers cues about the target semantics, the dimensionality mismatch between the textual and visual embeddings prevents direct alignment for this purpose. To find `<cls>` tokens that better represent target classes, we trace the historical `<cls>` tokens from target data streams with the lowest uncertainty across time steps, which we refer to as "anchors" capturing the context in the stream. As shown in Figure 2b, retaining these "anchors" follows a promising trajectory that gradually aligns with the text embeddings within the same visual space. These anchors serve as proxies to bridge visual and textual representations, guiding `<cls>` attentiveness to be more semantically aligned with target domains.

Building on these two insights, we design a context-aware token pruning and merging mechanism to condense tokens within the ViT blocks. As shown in Figure 1, the context from the target class is incorporated into the token condensation function via a context token reservoir (CTR). At each adaptation step, the most reliable `<cls>` tokens are retained as "anchors" and used to guide the selection and preservation of class-relevant tokens. These anchors, *in turn*, act as token-level classifiers to correct VLM predictions and improve the final visual-text alignment. Specifically, our token pruning strategy utilizes consensus across attention heads to retain only the most relevant class-specific patches, while class-ambiguous tokens are merged into representative centers using a coreset selection algorithm. Empirically, we show that a minimal number of centers is sufficient to stabilize performance, allowing TCA to scale efficiently with linear complexity.

To our knowledge, this is the first work to explore token condensation in the context of test-time adaptation. The efficiency and effectiveness of our training-free TCA approach have been rigorously validated across multiple TTA benchmark datasets, including ImageNet and four associated natural distribution shift datasets, along with ten fine-grained classification datasets. In extensive evaluations with traditional TTA baselines, prompting, and test-time prompting approaches, our method consistently outperforms the baselines by up to $21.4\%$ compared with the strongest baseline, while reducing GFLOPs cost by 12.2% to 48.9%. For reproducibility, we provide the complete benchmarking toolkit in the supplementary material for reference.

## 2 RELATED WORK

**Online Test-time Adaptation.** To address performance degradation during test time, online test-time adaptation (TTA) has gained significant attention. Current TTA methods can be categorized

into three main types (Wang et al., 2023c): optimization-based, data-based, and model-based approaches. Optimization-based methods focus on model updates and optimization objectives (Goyal et al., 2022; Marsden et al., 2024; Zhang et al., 2022). A prominent example is Tent (Wang et al., 2021), which adapts Batch Normalization layers (Ioffe & Szegedy, 2015) using entropy minimization. SAR (Niu et al., 2023) extends this approach to Layer Normalization (Ba et al., 2016) and Group Normalization (Wu & He, 2018) with sharpness-aware minimization (Foret et al., 2021). Data-based methods include augmentations like selective augmentation (Wang et al., 2022) and adversarial augmentation (Tomar et al., 2023), as well as the use of memory banks (Gong et al., 2022; Yuan et al., 2023; Chen et al., 2022). Model-based approaches involve architectural modifications to enhance model adaptability during testing (Liu et al., 2024a; Iwasawa & Matsuo, 2021; Jang et al., 2023; Wang et al., 2023b). However, these methods typically depend on large batch sizes and augmentations, which introduce significant latency for online prediction.

Recently, vision-language models like CLIP (Radford et al., 2021) have excelled beyond fixed label sets, rendering traditional TTA methods less suitable (Döbler et al., 2024). As a result, various online adaptation strategies have been proposed to improve zero-shot generalization. Test-time prompt tuning has emerged as a key approach in this context. TPT (Shu et al., 2022) optimizes learnable prompts using data augmentations and soft entropy minimization, Diff-TPT (Feng et al., 2023) enriches this with more diverse augmentations (Rombach et al., 2022), while C-TPT (Yoon et al., 2024) focusing on model calibration. Other methods like VTE (Döbler et al., 2024) and DART (Liu et al., 2024b) leverage prompt ensembles with DART further employing moving averages to boost performance. SwapPrompt (Ma et al., 2023) incorporates an EMA-updated target prompt. AdaPrompt (Zhang et al., 2024) utilizes a class-balanced memory bank to enhance adaptability. SCP (Wang et al., 2024) builds on TPT with a teacher-student framework to prevent semantic drift, while RLCF (Zhao et al., 2024) incorporates reinforcement learning strategy (Williams, 1992) to optimize the adaptation process. Beyond these, MTA (Zanella & Ayed, 2024) introduces a new objective based on test-time augmentation to optimize visual features in the semantic space. TDA (Karmanov et al., 2024) further improves CLIP's zero-shot ability by incorporating positive and negative caches with a training-free adapter. However, it relies on a large number of hyperparameters and is highly sensitive to them, while incurring significant computational costs during inference. In contrast, our approach strikes a better balance between computational efficiency and performance, outperforming both training-required and training-free methods.

**Token Condensation in Vision Transformers.** Vision transformers have achieved notable success in image recognition tasks, but their deployment is often limited by resource-constrained environments. To address this, various token condensation methods (Meng et al., 2022; Rao et al., 2021; Ryoo et al., 2021; Xu et al., 2022; Zong et al., 2022; Kong et al., 2022; Wang et al., 2023a) have been proposed to reduce the computational overhead, primarily through two strategies: token pruning and token merging. Token pruning eliminates less informative tokens to save computation, as seen in methods like EViT (Liang et al., 2022), which retains tokens based on their attentiveness to the `<cls>` tokens. ATS (Fayyaz et al., 2022) introduces input-dependent token pruning to adapt to variability across inputs. Token merging, on the other hand, seeks to combine similar tokens to reduce redundancy. For instance, ToME (Bolya et al., 2023) uses bipartite soft matching to merge neighboring tokens that exhibit similarity. Hybrid approaches have also emerged, such as TPS (Wei et al., 2023), which prunes tokens and transfers information to retained ones using nearest-neighbor matching, and PruMerge (Shang et al., 2024), which prunes inattentive tokens using interquartile range and merges via k-nearest neighbors. While previous works have focused on enhancing efficiency within pure ViT models, our approach utilizes token condensation from *a different perspective*: addressing multimodal distribution shifts in VLMs. This shift remains underexplored, particularly in how to use semantic guidance to prune irrelevant visual tokens that introduce ambiguity. By condensing these tokens, we effectively reduce such distribution shifts, enhancing test-time performance while simultaneously lowering computational costs.

## 3 OUR APPROACH

**Problem Set-up.** We begin by revisiting online test-time adaptation (TTA) of VLMs, focusing on contrastive language-image pre-training (CLIP) as a representative case. For a given downstream task $\mathcal{D}_{\text{tar}}$, the test data $\mathbf{x} = \{\mathbf{x}_t\}_{t=1}^T$ arrives sequentially at each time step $t$. The objective is to adapt CLIP on the fly to classify the incoming test samples into one of $C$ classes, each represented by a textual prompt like "a photo of a `<classname>`". CLIP embeds both visual and textual

inputs into a shared space. The visual encoder $E_v$ extracts visual features $\mathbf{z}_t = E_v(\mathbf{V}_t) \in \mathbb{R}^D$ from image patches $\mathbf{V}_t = [\mathbf{v}_{\text{cls}}, \mathbf{v}_1, \ldots, \mathbf{v}_N] \in \mathbb{R}^{(N+1) \times D_v}$ of dimension $D_v$, where $\mathbf{v}_{\text{cls}}$ is a special `<cls>` token appended to $N$ patches. The text encoder $E_t$ generates class embeddings $\mathbf{T} = \{\mathbf{t}_c\}_{c=1}^C$, where each $\mathbf{t}_c \in \mathbb{R}^D$ corresponding to a class prompt. Classification is performed by computing the cosine similarity between the visual embedding $\mathbf{z}_t$ and each class embedding $\mathbf{t}_c$ with the probabilities calculated as:

$$\mathbf{p}_{t,c}(\mathbf{z}_t, \mathbf{t}_c) = \frac{\exp\left(\cos(\mathbf{z}_t, \mathbf{t}_c)/\tau\right)}{\sum_{j=1}^C \exp\left(\cos(\mathbf{z}_t, \mathbf{t}_j)/\tau\right)}, \tag{1}$$

where $\tau$ denotes the temperature parameter controlling the sharpness of the output distribution.

**Pitsfalls of TTA.** Since the target domain $\mathcal{D}_{\text{tar}}$ is unseen during CLIP's pre-training, the alignment between visual embeddings $\mathbf{z}_t$ and the textual embeddings $\mathbf{T}$ may be suboptimal. Previous methods have attempted to address this by learning domain-specific prompts (Yoon et al., 2024) or replacing classifier weights with visual centroids (Iwasawa & Matsuo, 2021) to move $\mathbf{T}$ closer to $\mathbf{z}_t$. However, the variability in CLIP's visual embeddings is often much *higher* than in textual embeddings (Radford et al., 2021). At the patch level, individual tokens within the visual embeddings can drift and vary significantly (Radford et al., 2021). Thus, it becomes more urgent to derive methods that adjust $\mathbf{z}_t$ towards $\mathbf{T}$ for improved alignment.

**Do Visual Tokens Correlate to Drift?** In VLMs, the visual encoder $E_v$ is typically a Vision Transformer (ViT), where patch tokens play a key role in forming the visual representation. To investigate the sources of *visual shifts*, we first analyze the roles of individual patch tokens in the misalignment of text embeddings. Using a ***leave-one-out*** strategy, we sequentially remove each token based on its attentiveness to the `<cls>` token and measure the resulting impact on the similarity of $\mathbf{z}_t$ to its corresponding text embedding $\mathbf{t}_c$ (see Figure 2a). Our analysis reveals that token contributions to the final prediction are *uneven*—removing tokens with lower correlations to `<cls>` (highlighted in grey and blue) often leads to no decrease in performance, or even slightly improves alignment. This observation suggests that *class-ambiguous* or *class-irrelevant* token groups introduce noise or drift in the visual representation, highlighting the need and feasibility for strategies to manipulate these tokens during adaptation.

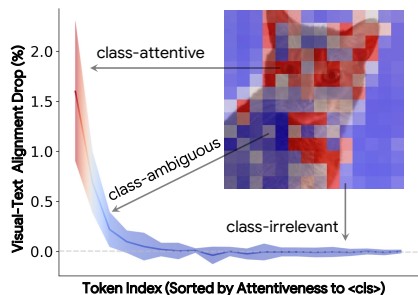

(a) Impact of Token Removal to Alignment

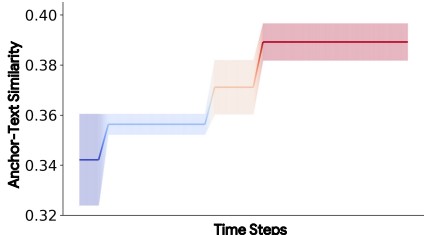

(b) Anchor-Text Alignment Over Time

Figure 2: Preliminary studies of token influence and anchor strategy.

**How to Mine `<cls>` Tokens Aligned with Text?** In CLIP pretraining, the `<cls>` token is trained to align with a vast array of concepts, leading to attentiveness patterns that may extend *beyond* the target classes. Given the mismatch in dimensionality between textual and visual tokens, a fundamental challenge remains in finding a more representative `<cls>` token that consistently aligns with the text embeddings. To address this, we exploit temporal cues by tracing target samples with the lowest uncertainty as "anchors". The similarity between these anchors and $\mathbf{T}$ is plotted in Figure 2b. Our findings indicate that retaining these "anchors" with minimal uncertainty leads to a steady convergence of visual embeddings with the corresponding text embeddings in the same visual space. These anchors serve as contextually relevant proxies, effectively bridging the gap between visual and textual representations and enabling `<cls>` attentiveness to align more consistently with target classes.

### 3.1 TOKEN CONDENSATION AS ADAPTATION

Building upon our empirical findings, we propose a novel strategy called **Token Condensation as Adaptation (TCA)**, which selectively removes or merges tokens contributing to drift, enabling efficient adaptation in a training-free manner. Specifically, given an $L$-layer ViT, the forward pass

through the $l$-th Transformer block, where $l \in [1, 2, \ldots, L]$, is formulated as:

$$\mathbf{V}^{l+1} = \hat{\mathbf{V}}^l + \text{MLP}(\hat{\mathbf{V}}^l), \hat{\mathbf{V}}^l = \mathbf{V}^l + \frac{1}{H} \sum_{h=1}^{H} \text{Attention}(\mathbf{V}^l \mathbf{W}_Q^h, \mathbf{V}^l \mathbf{W}_K^h) \mathbf{V}^l \mathbf{W}_V^h, \quad (2)$$

where $\mathbf{V}^l \in \mathbb{R}^{(N+1) \times D_v}$ represents the token embeddings at layer $l$. The matrices $\mathbf{W}_Q^h$, $\mathbf{W}_K^h$, $\mathbf{W}_V^h \in \mathbb{R}^{D_v \times D_v}$ represent linear projection for the query, key, and value vectors in the $h$-th attention head, and $H$ denotes the total number of attention heads. To effectively condense tokens without compromising performance, we introduce two key functions: the token pruning function $f_{\text{prune}}$ and the token merging function $f_{\text{merge}}$. In line with (Liang et al., 2022), these functions are applied between the multi-head self-attention layers and feed-forward layers of each transformer block. The modified forward pass becomes:

$$\hat{\mathbf{V}}^l = f_{\text{merge}} \circ f_{\text{prune}} \left( \mathbf{V}^l; \mathfrak{R} \right), \quad (3)$$

where $f_{\text{prune}}(\cdot; \mathfrak{R}) : \mathbb{R}^{N+1} \mapsto \mathbb{R}^{(\alpha \cdot R \cdot N)+1}$ and $f_{\text{merge}}(\cdot; \mathfrak{R}) : \mathbb{R}^{(\alpha \cdot R \cdot N)+1} \mapsto \mathbb{R}^{R \cdot N+1}$ are responsible for reducing the number of tokens from $N + 1$ (including the `<cls>` token) to $R \cdot N + 1$, where $R$ represents the fraction of tokens to be preserved. The parameter $\alpha$ controls the extent of token pruning, ensuring only the most semantically significant tokens are retained. Detailed explanation of $f_{\text{prune}}$ and $f_{\text{merge}}$ can be found in Section 3.2 and 3.3.

**Context Token Reservoir $\mathfrak{R}$.** Notably, both $f_{\text{prune}}$ and $f_{\text{merge}}$ rely on a class-specific token reservoir $\mathfrak{R} = \{\mathfrak{R}_c\}_{c=1}^C$. Each buffer $\mathfrak{R}_c = \{(\mathbf{H}_c(\mathbf{z}_i; \mathbf{t}_c), \mathbf{A}_{i,c}^{\text{cls}})\}_{i=1}^M$ is structured as a *priority queue* that retains the top $M$ most reliable anchor target samples, which serve to implicitly distil semantic information from the corresponding text prompt $\mathbf{t}_c$ to guide the visual adaptation. These anchors are crucial *alignment proxies*: although the architectures of the text encoder $E_t$ and the visual encoder $E_v$ differ, the selected anchor samples help determine which visual tokens best align with text features from CLIP's perspective. The reliability of these anchors is quantified by entropy scores $\mathbf{H}_c(\mathbf{z}_t, \mathbf{t}_c)$,

$$\mathbf{H}_c(\mathbf{z}_t, \mathbf{t}_c) = -\mathbf{p}_{t,c}(\mathbf{z}_t, \mathbf{t}_c) \log \mathbf{p}_{t,c}(\mathbf{z}_t, \mathbf{t}_c), \quad (4)$$

which act as keys to update the reservoir $\mathfrak{R}_c$. At each time step $t$, for each visual embedding $\mathbf{z}_t$, the corresponding `<cls>` embeddings from all $L$ layers $\mathbf{A}_{t,c}^{\text{cls}} = [\mathbf{v}_{\text{cls}}^1, \ldots, \mathbf{v}_{\text{cls}}^L] \in \mathbb{R}^{L \times D_v}$ will be stored in $\mathfrak{R}_c$ if $\text{argmax}(\mathbf{p}_{t,c}) = c$, ensuring that only the most semantically consistent samples are retained:

$$\mathfrak{R}_c \leftarrow \text{update}\left(\mathfrak{R}_c, \left(\mathbf{H}_c(\mathbf{z}_t, \mathbf{t}_c), \mathbf{A}_{t,c}^{\text{cls}}\right)\right). \quad (5)$$

If the priority queue $\mathfrak{R}_c$ has reached its capacity $M$, the sample with the highest entropy score is discarded, and the new sample is inserted. Strategies for *updating* the reservoir such as first-in, first-out (FIFO) policies and similarity- and diversity-enforcing methods are explored in Section 4.3.

**Logits Self-correction.** To counter the shifts on the *semantic* side, we introduce a logits self-correction mechanism that leverages anchor tokens stored in $\mathfrak{R}$. In particular, the `<cls>` embedding of the current sample, $\mathbf{V}_t^{\text{cls}} \in \mathbb{R}^{L \times D_v}$, is compared with the stored anchors denoted as a set $\mathcal{A} = \{\mathbf{A}_{i,c}^{\text{cls}}\}_{i=1}^M$. The cosine similarity between these cross-layer `<cls>` tokens serves as a token-level classifier, which provides auxiliary information to adjust the predicted probability $\mathbf{p}_{t,c}$ from a *visual* perspective:

$$\tilde{\mathbf{p}}_{t,c} = \mathbf{p}_{t,c} + \lambda \mathbf{p}_{t,c}^{\text{token}}, \quad \mathbf{p}_{t,c}^{\text{token}} = \frac{1}{M} \sum_{i=1}^{M} \cos(\mathbf{V}_t^{\text{cls}}, \mathbf{A}_{i,c}^{\text{cls}}) \cdot \mathbf{P} \cdot \mathbb{1}_c, \quad (6)$$

where $\lambda$ is the logit correction weight and $\mathbb{1}_c \in \mathbb{R}^C$ the one-hot vector for the $c$-the class. The layer-specific exponential scaling coefficients are denoted as $\mathbf{P} = [\exp(\frac{l}{\beta})]_{l=1}^L \in \mathbb{R}^L$, where $\beta$ controls the influence of different layers. We show that this correction temperature $\beta$ provides semantic interpretability, as further discussed in Section 4.3. This self-correction mechanism ensures that the final prediction is better aligned with the visual and semantic contexts, improving robustness in handling semantic shifts during inference.

## 3.2 CONTEXT-AWARE CROSS-HEAD TOKEN PRUNING $f_{\text{prune}}$

Prior token pruning methods for ViTs such as (Liang et al., 2022) primarily discard patch tokens with lower averaged attention scores $\mathbf{S} \in \mathbb{R}^N$ relative to the `<cls>` token $\mathbf{v}_{\text{cls}}^l$ across all attention

heads. The pruning process can be expressed as

$$\hat{\mathbf{V}}^l_{\text{prune}} \leftarrow \{\hat{\mathbf{v}}^l_i \mid \mathbf{S}_i \leq \theta_{\text{prune}}(\alpha, R), \forall i \in [N]\}, \ \mathbf{S}_i = \frac{1}{H}\sum_{h=1}^{H}\text{Attention}(\mathbf{v}^l_{\text{cls}}\mathbf{W}^h_Q, \mathbf{v}^l_i \mathbf{W}^h_K), \quad (7)$$

where $\hat{\mathbf{V}}^l_{\text{prune}}$ denotes the set of tokens retained after pruning at layer $l$. Here, the pruning threshold $\theta_{\text{prune}}(\alpha, R)$ is determined by the desired pruning ratio, ensuring only top-ranked $\alpha \cdot R \cdot N$ tokens are retained. However, this approach faces two limitations when applied in TTA tasks: (1) The `<cls>` token is *universal* and may not be specifically aligned with the target class set. It may capture broad,

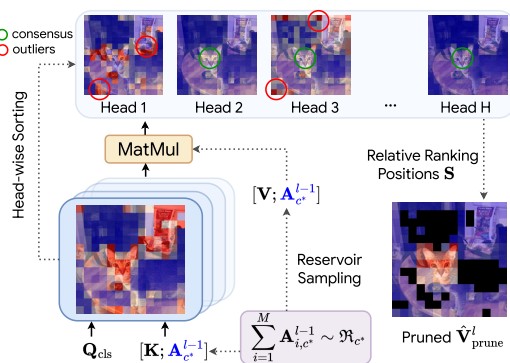

Figure 3: An illustration of context-aware cross-head token pruning.

unrelated semantics (*e.g.,* "cat food"), leading to the retention of irrelevant tokens that mislead the model into making incorrect predictions of the target class (*e.g.,* "cat"). (2) Averaging attention scores across all heads risks omitting important details, as each attention head tends to focus on distinct features (*e.g.,* shape, color). Outliers in attention heads (highlighted by red circles in Figure 3) may disproportionately dominate the overall score, overshadowing valuable information by other heads and leading to suboptimal pruning decisions. To overcome these limitations, we propose a *cross-head token pruning* function that evaluates the token importance individually for each attention head and utilizes the averaged relative ranking positions to determine which tokens to prune (see Figure 3). This approach reaches a more robust cross-head consensus and mitigates the impact of outliers. To guide pruning with more representative `<cls>` tokens, we sample an anchor `<cls>` token for the $(l-1)$-th layer from $\mathfrak{R}_{c^*}$, where $c^*$ is determined by comparing the cosine similarity between the current $\mathbf{v}^l_{\text{cls}}$ token embedding and the stored anchor tokens $\mathbf{A}^{l-1}_{i,c} \in \mathcal{A}$:

$$\mathbf{A}^{l-1}_{c^*} = \frac{1}{M}\sum_{i \in [M]} \mathbf{A}^{l-1}_{i,c^*}, \ c^* = \arg\max_{c \in [C]} \cos(\mathbf{v}^l_{\text{cls}}, \mathbf{A}^{l-1}_{i,c}), \quad (8)$$

where $\mathbf{A}^{l-1}_{c^*} \in \mathbb{R}^{D_v}$ is the averaged anchor token sampled from $\mathfrak{R}_{c^*}$, providing historical context that is *better* aligned with target semantics. Subsequently, we refine the attention map by inserting this historical anchor $\mathbf{A}^{l-1}_{c^*}$ to compute the pruning scores $\mathbf{S}^{\text{head}}_i$ for the $i$-th token:

$$\mathbf{S}^{\text{head}}_i = \frac{1}{H}\sum_{h=1}^{H}\text{rank}_h(i), \ \hat{\mathbf{V}}^l_{\text{prune}} \leftarrow \{\hat{\mathbf{v}}^l_i \mid \mathbf{S}^{\text{head}}_i \leq \theta_{\text{prune}}(\alpha, R), \forall i \in [N]\}, \quad (9)$$

$$\text{rank}_h(i) = \text{argsort}(\text{Attention}([\mathbf{v}^l_{\text{cls}}; \mathbf{A}^{l-1}_{c^*}]\mathbf{W}^h_Q, [\mathbf{V}^l; \mathbf{A}^{l-1}_{c^*}]\mathbf{W}^h_K),$$

where $[\cdot; \cdot]$ indicates concatenation and $\text{rank}_h(i)$ gives the relative ranking positions of token $i$ in head $h$ based on its attention score to $\mathbf{v}^l_{\text{cls}}$ and $\mathbf{A}^{l-1}_{c^*}$. This method ensures that tokens receiving consistently high attention across individual heads are retained, thereby achieving greater robustness to outliers in specific attention heads.

### 3.3 CONTEXT-AWARE TOKEN MERGING $f_{\text{merge}}$

As depicted in Figure 2a, a subset of tokens, although relevant to the target class, exhibit high uncertainty. These tokens are referred to as *class-ambiguous* tokens, identified from the ranked token list derived using Equation (9):

$$\Phi = \{i \mid \theta_{\text{merge}}(R) \leq \mathbf{S}^{\text{head}}_i \leq \theta_{\text{prune}}(\alpha, R), \forall i\}, \quad (10)$$

where $\theta_{\text{merge}}(R)$ denotes thresholds for token selection during merging. The selected tokens $\mathbf{V}^l_{\Phi} = \{\mathbf{v}^l_i\}_{i \in \Phi}$ can introduce variance or noise into latent representation $\mathbf{z}_t$ and negatively impact the final classification decision. To address this, we propose a *context-aware token merging strategy* to consolidate these tokens into more representative ones.

Existing token merging strategies either truncate neighbored token pairs with high similarity with bipartite soft matching (Bolya et al., 2023) or apply spectral clustering (Bianchi et al., 2020), and graph pooling (Wu et al., 2022) to merge similar tokens at higher costs. In contrast, we adopt a more efficient *coreset selection* approach, which identifies the most representative tokens $\hat{\mathbf{V}}^l_{\text{merge}} \in \mathbb{R}^{K \times D_v}$ from $\mathbf{V}^l_{\Phi}$ and assigns the remaining ambiguous tokens to these selected tokens. The coreset selection strategy is equivalent to solving the K-Center problem (Wolf, 2011; Sener & Savarese, 2018). The objective is to select $K$ center tokens such that the maximum distance between any token and its nearest center is minimized. Formally, the greedy search for optimization is defined as follows:

$$\mathbf{C}^* = \underset{\mathbf{C} \subseteq \mathbf{V}^l_{\Phi}, |\mathbf{C}| = K}{\arg\min} \max_{\mathbf{v}^l_i \in \mathbf{V}^l_{\Phi}} \min_{\mathbf{v}^l_c \in \mathbf{C}} d(\mathbf{v}^l_i, \mathbf{v}^l_c), \tag{11}$$

where $\mathbf{C}^* \in \mathbb{R}^{K \times D_v}$ represents the set of selected center tokens, $K$ is the number of centers, and $d(\cdot, \cdot)$ is the distance metric between token $\mathbf{v}^l_i$ and center token $\mathbf{v}^l_c$. Once the center tokens $\mathbf{C}^*$ are selected, the remaining tokens are assigned to their nearest centers, and the ambiguous tokens are merged as:

$$\hat{\mathbf{V}}^l_{\text{merged}} = \frac{1}{|\mathcal{N}(k)|} \sum_{\mathbf{v}^l_i \in \mathcal{N}(k)} \mathbf{v}^l_i, \tag{12}$$

where $\mathcal{N}(k)$ represents the set of tokens assigned to center $k$. As demonstrated in Section A.1, the value of $K$ can be kept small, with $K \ll N$, allowing our merging algorithm to operate with linear complexity. The final embedding is composed of the class token $\mathbf{v}^l_{\text{cls}}$, the pruned tokens $\hat{\mathbf{V}}^l_{\text{prune}}$ excluding those in the class-ambiguous set $\Phi$, and the merged tokens $\hat{\mathbf{V}}^l_{\text{merge}}$. This combined embedding is then passed to the next layer in the ViT. The algorithm can be found via Algorithm 1.

# 4 EXPERIMENTS

## 4.1 EXPERIMENTAL SETUP

**Datasets.** Following prior works, we conducted two main benchmarks: *the cross-dataset (CD) benchmark* and *the out-of-distribution (OOD) benchmark*. The CD benchmark assesses the model's performance on unseen classes across 10 datasets: Aircraft (Maji et al., 2013), Caltech101 (Fei-Fei et al., 2007), Cars (Krause et al., 2013), DTD (Cimpoi et al., 2014), EuroSAT (Helber et al., 2019), Flower102 (Nilsback & Zisserman, 2008), Food101 (Bossard et al., 2014), Pets (Parkhi et al., 2012), SUN397 (Xiao et al., 2010), and UCF101 (Soomro et al., 2012). In contrast, the OOD benchmark focuses on evaluating the model's effectiveness on shifted data using label sets previously seen by CLIP. This includes variants of ImageNet (Deng et al., 2009): ImageNet-A (Hendrycks et al., 2019), ImageNet-V2 (Recht et al., 2019), ImageNet-R (Hendrycks et al., 2021), and ImageNet-S (Wang et al., 2019).

**Baselines.** To provide a comprehensive evaluation, we compare TCA with existing approaches across four categories: (1) *Prompt-tuning methods* like CoOp (Zhou et al., 2022b) and CoCoOp (Zhou et al., 2022a), which require multi-epoch adaptation; (2) *Conventional online test-time adaptation (TTA) methods* such as Tent (Wang et al., 2021) and SAR (Niu et al., 2023). Tent updates batch normalization layers, while SAR further incorporates sharpness-aware minimization for reliable model updates. Following Döbler et al. (2024), we reran these experiments with adjusted batch sizes to align with our settings; (3) *Test-time prompting methods*, including TPT (Shu et al., 2022), C-TPT (Yoon et al., 2024), and Diff-TPT (Feng et al., 2023), as well as TTA methods for CLIP such as MTA (Zanella & Ayed, 2024) and TDA (Karmanov et al., 2024); and (4) *Token pruning and merging methods for ViTs*, such as EViT (Liang et al., 2022), ToMe (Bolya et al., 2023), and ATS (Fayyaz et al., 2022). As ATS is an adaptive token pruning method with no fixed budget, we constrain its computational cost by an upper bound to ensure fair comparison.

**Implementation Details.** We utilize the official CLIP [1] prompts as text inputs. The batch size is set to 1 *without* data augmentations to mimic realistic deployment scenarios. All experiments are conducted using the pre-trained CLIP models, specifically using ViT-B/16 and ViT-L/14 architectures as the visual backbone. For both CD and OOD benchmarks, we set $K$ to 2. Notably, our method is training-free, which achieves rapid adaptation with no need for any hyperparameters for optimization. All experiments are performed on a single NVIDIA RTX A6000 GPU.

---

[1] https://github.com/openai/CLIP

Table 1: Results on the cross-dataset benchmark using CLIP ViT-B/16, including the number of learnable parameters (L-Param.) for learning-based TTA methods. * denotes the averaged GFLOPs across all datasets.

| Method | Aug-free | Aircraft | Caltech101 | Cars | DTD | EuroSAT | Flower102 | Food101 | Pets | SUN397 | UCF101 | Average | GFLOPs | L-Param. |
|---|---|---|---|---|---|---|---|---|---|---|---|---|---|---|
| CLIP | ✓ | 23.22 | 93.55 | 66.11 | 45.04 | 50.42 | 66.99 | 82.86 | 86.92 | 65.63 | 65.16 | 64.59 | 17.59 | 0 |
| CoOp | ✗ | 18.47 | 93.70 | 64.51 | 41.92 | 46.39 | 68.71 | 85.30 | 89.14 | 64.15 | 66.55 | 63.88 | 17.59 | 2048 |
| CoCoOp | ✗ | 22.29 | 93.79 | 64.90 | 45.45 | 39.23 | 70.85 | 83.97 | 90.46 | 66.89 | 68.44 | 64.63 | 17.59 | 34,816 |
| Tent | ✓ | 8.97 | 93.39 | 62.69 | 39.78 | 20.85 | 61.23 | 83.70 | 87.76 | 65.30 | 66.93 | 59.06 | 17.59 | 40,960 |
| SAR | ✓ | 21.09 | 91.85 | 61.15 | 44.68 | 46.19 | 63.54 | 81.43 | 87.95 | 59.74 | 65.58 | 62.32 | 17.59 | 31,744 |
| TPT | ✗ | 24.78 | 94.16 | 66.87 | 47.75 | 42.44 | 68.98 | 84.67 | 87.79 | 65.50 | 68.04 | 65.10 | 1108.61 | 2048 |
| Diff-TPT | ✗ | 25.60 | 92.49 | 67.01 | 47.00 | 43.13 | 70.10 | 87.23 | 88.22 | 65.74 | 62.67 | 65.47 | - | - |
| C-TPT | ✗ | 23.90 | 94.10 | 66.70 | 46.80 | 48.70 | 69.90 | 84.50 | 87.40 | 66.00 | 66.70 | 65.47 | 1108.61 | 2048 |
| MTA | ✗ | 25.32 | 94.21 | 68.47 | 45.90 | 45.36 | 68.06 | 85.00 | 88.24 | 66.67 | 68.11 | 65.53 | - | - |
| TDA | ✓ | 23.91 | 94.24 | 67.28 | 47.40 | 58.00 | 71.42 | 86.14 | 88.63 | 67.62 | 67.53 | 67.53 | 17.59 | 0 |
| EViT$_{R=0.9}$ | ✓ | 24.12 | 92.25 | 64.57 | 45.09 | 48.41 | 70.24 | 84.99 | 88.96 | 64.58 | 68.46 | 65.17 | 15.41 | 0 |
| ToME$_{R=0.9}$ | ✓ | 24.66 | 92.49 | 63.10 | 44.92 | 48.64 | 69.22 | 85.04 | 87.90 | 64.22 | 68.62 | 64.88 | 15.31 | 0 |
| ATS$_{R=0.9}$ | ✓ | 22.86 | 92.21 | 57.90 | 40.96 | 40.62 | 67.52 | 80.16 | 85.34 | 61.53 | 67.22 | 61.63 | 11.15* | 0 |
| EViT$_{R=0.7}$ | ✓ | 23.31 | 91.20 | 58.44 | 43.32 | 43.26 | 67.11 | 79.70 | 85.77 | 61.41 | 66.69 | 62.02 | 11.62 | 0 |
| ToME$_{R=0.7}$ | ✓ | 22.26 | 90.79 | 55.48 | 42.32 | 40.12 | 64.11 | 79.36 | 84.19 | 60.66 | 63.97 | 60.33 | 11.45 | 0 |
| ATS$_{R=0.7}$ | ✓ | 17.28 | 85.40 | 33.65 | 36.52 | 27.79 | 52.62 | 55.97 | 72.94 | 48.82 | 56.44 | 48.74 | 8.76* | 0 |
| **TCA**$_{R=0.9}$ | ✓ | 24.87 | 93.63 | 65.33 | 46.16 | 70.43 | 73.33 | 85.31 | 89.53 | 65.92 | 72.38 | 68.69 | **15.45**$_{-12.2\%}$ | 0 |
| **TCA**$_{R=0.7}$ | ✓ | 23.19 | 92.13 | 58.15 | 44.50 | 61.63 | 69.79 | 79.99 | 85.99 | 61.89 | 67.38 | 64.46 | **11.69**$_{-33.5\%}$ | 0 |

## 4.2 COMPARISON WITH STATE-OF-THE-ART

### 4.2.1 CROSS-DATASET BENCHMARK

Table 1 presents the results for fine-grained cross-dataset benchmark using the ViT-B/16 architecture. As observed in Figure 2a, the core idea behind TCA is that condensing inattentive tokens can effectively mitigate distribution shifts caused by visual-text misalignment. This concept is first validated by the improved performance of token pruning baselines over CLIP inference, where a condensed token set yields a $0.9\%$ increase in average accuracy when $R = 0.9$. TCA further enhances its performance by dealing with visual-text misalignment, moving visual features toward historical anchor tokens from CTR. As a result, TCA achieves an average accuracy of 68.69%, outperforming both train-required and training-free baselines without augmentation. Conventional TTA methods perform poorly on all datasets even with the requirement of fine-tuning a large amount of learnable parameters. In contrast, prompt-tuning methods, although requiring fewer learnable parameters, rely heavily on augmentation and struggle to effectively handle visual shifts. While TDA is a training-free method, it requires a large number of hyperparameters (a total of 10 for managing positive and negative caches) to achieve optimal performance. On the other hand, TCA uses significantly fewer hyperparameters and delivers a 1.72% improvement in average accuracy over TDA, with approximately 12.2% fewer GFLOPs. Further details on TDA combined with token condensation baselines can be found in Figure 4c. To verify the universality of the proposed TCA, we examine the impact of the visual backbone (ViT-L/14), where we reduce 48.9% GFLOPs without compromising adaptation. The results are presented in Section A.1.

### 4.2.2 OUT-OF-DISTRIBUTION BENCHMARK

A consistent observation can be seen in the out-of-distribution (OOD) benchmark, where TCA demonstrates significant improvements over the CLIP baseline under a constrained GFLOPs budget of $R = 0.95$, as shown in Table 3. TCA outperforms traditional test-time adaptation methods while maintaining efficiency. TCA also achieves superior results on ImageNet-R and ImageNet-S, outperforming TPT without augmentation. Additionally, when compared to other training-based approaches, even those with unlimited computational budgets, TCA delivers comparable performance. However, we observe that TCA does not perform as strongly on the OOD benchmark as it does on the CD benchmark even with a higher rate $R$. This may be due to the conceptual shifts in OOD datasets, as shown in Section A.4, which could present a challenge for training-free adaptation methods.

## 4.3 ABLATION STUDY

We conducted a comprehensive ablation study to evaluate TCA's effectiveness and efficiency. For further analysis on reservoir size, merging center, pruning to merging ratio, visual backbone, and more baseline comparisons, see Section A.1.

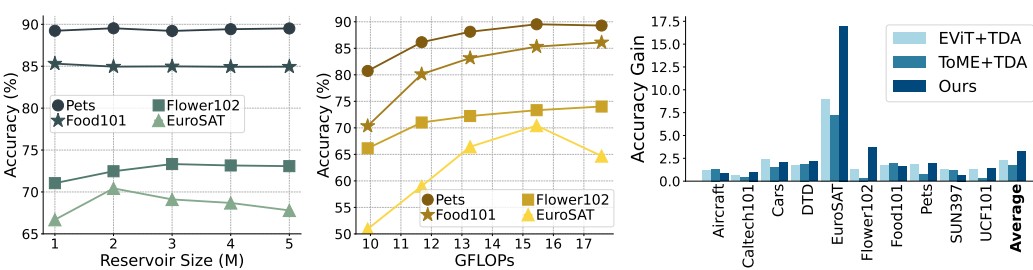

(a) Impact of reservoir size.    (b) Impact of GFLOPs budget.  (c) TCA v.s. TDA with token condensation.

Figure 4: Impact of TCA on performance under different configurations.

Table 3: Results on the out-of-distribution benchmark with CLIP ViT-B/16. $*$ denotes the averaged GFLOPs across all datasets.

| Method | Aug-free | ImageNet | ImageNet-A | ImageNet-V2 | ImageNet-R | ImageNet-S | **Average** | **OOD Average** | **GFLOPs** |
|---|---|---|---|---|---|---|---|---|---|
| CLIP | ✓ | 68.34 | 49.89 | 61.88 | 77.65 | 48.24 | 61.20 | 59.42 | 17.59 |
| Tent | ✓ | 65.49 | 44.57 | 59.26 | 78.72 | 22.52 | 54.11 | 51.27 | 17.59 |
| SAR | ✓ | 58.52 | 33.71 | 53.95 | 76.08 | 39.24 | 52.30 | 50.75 | 17.59 |
| TPT | ✗ | 68.98 | 54.77 | 63.45 | 77.06 | 47.94 | 62.44 | 60.81 | 1108.61 |
| Diff-TPT | ✗ | 70.30 | 55.68 | 65.10 | 75.00 | 46.80 | 62.28 | 60.52 | - |
| C-TPT | ✗ | 69.30 | 52.90 | 63.40 | 78.00 | 48.50 | 62.42 | 60.70 | 1108.61 |
| MTA | ✗ | 70.08 | 58.06 | 64.24 | 78.33 | 49.61 | 64.06 | 62.56 | - |
| TDA | ✓ | 69.26 | 50.82 | 62.23 | 77.93 | 50.26 | 62.10 | 60.31 | 17.59 |
| EViT$_{R=0.95}$ | ✓ | 68.32 | 49.46 | 61.73 | 77.00 | 47.76 | 60.85 | 58.99 | 16.31 |
| ToME$_{R=0.95}$ | ✓ | 67.57 | 48.81 | 60.88 | 75.78 | 47.05 | 60.02 | 58.13 | 16.21 |
| ATS$_{R=0.95}$ | ✓ | 65.83 | 49.80 | 59.47 | 71.09 | 43.38 | 57.91 | 55.94 | 11.50$*$ |
| **TCA**$_{R=0.95}$ | ✓ | 68.88 | 50.13 | 62.10 | 77.11 | 48.95 | 61.43 | 59.57 | 16.55 |

**Impact of Logits Correction Temperature $\beta$.** In Table 2, we examine how different logits correction temperatures $\beta$ affect the adaptation results. The intuition is that with a smaller $\beta$ value, the logits correction will emphasize the tokens in shallower layers (Equation (6)), while a

Table 2: Impact of scale factor $\beta$.

| $\beta$ | **0.01** | **0.05** | **1** | **3** | **5** |
|---|---|---|---|---|---|
| Pets | 89.51 | **89.53** | 89.37 | 89.42 | 89.26 |
| Flower102 | **73.33** | 73.08 | 70.93 | 70.56 | 70.44 |
| EuroSAT | 63.64 | 64.06 | 69.86 | 70.26 | **70.43** |

larger $\beta$ value will shift the focus to deeper layers. We observe that a smaller value of $\beta$ is preferred for the Pets dataset as it contains animals as objects, requiring more high-level contextual information for accurate predictions (Raghu et al., 2021). In contrast, for EuroSAT, the best predictions are obtained with larger $\beta$ values, suggesting that low-level, local information is crucial. This aligns well with the nature of the dataset, where different types of land can be distinguished by features such as colors and edges. Nevertheless, our method consistently provides significant improvements across all $\beta$ values, with accuracy gains of up to 20%, highlighting the effectiveness of logits correction using the anchor tokens.

**Impact of Correction Weight $\lambda$.** To investigate how different correction weights $\lambda$ affect performance, as described in Equation (6), we conducted experiments across a wide range of $\lambda$ values, from 2 to 8, as shown in Table 4. We observe that Pets exhibits stable results across

Table 4: Impact of correction weight $\lambda$.

| $\lambda$ | **2** | **3** | **4** | **5** | **6** | **7** | **8** |
|---|---|---|---|---|---|---|---|
| Pets | **89.53** | 89.32 | 89.13 | 88.96 | 88.96 | 88.66 | 88.44 |
| Flower102 | 72.43 | 72.76 | 73.20 | 73.16 | 73.16 | **73.33** | 73.16 |
| EuroSAT | 60.15 | 65.74 | 68.80 | 69.51 | 69.84 | 70.16 | **70.43** |

different $\lambda$ values, indicating that less aggressive correction is sufficient. In contrast, datasets such as Flower102 and EuroSAT which initially do not perform well on CLIP, benefit from stronger corrections, achieving their best performance with larger correction weights of 7 and 8, respectively. This highlights the effectiveness of our logits correction module.

**Impact of GFLOPs Budget.** We evaluate TCA's performance under different GFLOPs budgets: $R = \{0.6, 0.7, 0.8, 0.9\}$, resulting in GFLOPs of 9.91, 11.68, 13.27, and 15.45, respectively, compared to the baseline ($R = 1$, 17.58 GFLOPs). As shown in Figure 4b, condensing inattentive tokens can even enhance performance on certain datasets, notably Pets, and EuroSAT. Specifically for EuroSAT, when $R = 0.9$, the model's adaptation performance is significantly improved, aligning with our findings in Figure 2a. However, excessively aggressive pruning budgets (*e.g.,* GFLOPs less than 13) lead to significant performance degradation across all datasets. This occurs since higher prun-

ing rates may inadvertently remove informative tokens, causing irreversible harm in training-free scenarios where we lack supervision or the ability to update the model for extra correction.

**Impact of Reservoir Saving Strategy.** In Table 5, we examine the performance changes across different reservoir saving strategies. We compare several approaches: *First-In, First-Out (FIFO)*; an *uncertainty-based* strategy, which discards the most uncertain sample when the reservoir reaches capacity; a *similarity-enforced* strategy, where samples with high certainty and high cosine similarity to the saved samples are preferred; and a *diversity-enforced* strategy, which prioritizes saving prototypes that contain distinct tokens compared to those already stored.

Table 5: Impact of reservoir updating strategy.

|          | FIFO  | Uncertainty | Similarity-enf | Diversity-enf |
| -------- | ----- | ----------- | -------------- | ------------- |
| Pets     | 89.21 | 89.18       | 88.91          | **89.53**     |
| Flower102| 70.65 | 73.28       | 72.15          | **73.33**     |
| EuroSAT  | 50.28 | 70.20       | 68.48          | **70.43**     |

Our results show that the FIFO strategy performs poorly on Flower102 and EuroSAT, likely because CLIP's low confidence leads to retaining misclassified samples. Conversely, Pets has high CLIP zero-shot accuracy (86.91% in Table 1), which makes FIFO acceptable. Among all strategies, the diversity-based approach consistently achieves the best performance. This is intuitive, as it maintains a representative set of features by capturing dataset diversity, whereas entropy-based methods may store homogenous features and overlook multiple class prototypes. By prioritizing diversity, our method ensures that a more representative set of features is maintained, leading to more robust performance across datasets.

Table 6: Impact of component.

| $A_{c*}^l$ | $S^{head}$ | Food101 | Pets  | EuroSAT |
| ---------- | ---------- | ------- | ----- | ------- |
|            |            | 85.25   | 88.77 | 68.14   |
| ✓          |            | 85.31   | 88.96 | 67.69   |
|            | ✓          | 85.25   | 89.23 | 67.14   |
| ✓          | ✓          | **85.31** | **89.53** | **70.43** |

**Impact of Component.** The impact of the historical anchor $A_{c*}^l$ and the head-wise sorting score $S^{head}$ (Equation (8)) is presented in Table 6. We observe that each component individually contributes to performance improvements. On the Food101 and Pets datasets, incorporating either component yields measurable gains in accuracy. By leveraging historical anchors, the model acquires rich contextual information, enhancing the stability of token importance over time. Simultaneously, cross-head token sorting ensures that token pruning decisions are more robust by accounting for consensus across attention heads. An intriguing case arises with the EuroSAT dataset. Here, the baseline performance without any components is 68.14%. Applying either component alone results in a slight performance decrease. However, when both components are used together, performance significantly improves to 70.43%. This outcome emphasizes the necessity of combining historical anchors and cross-head token sorting to fully realize the model's potential.

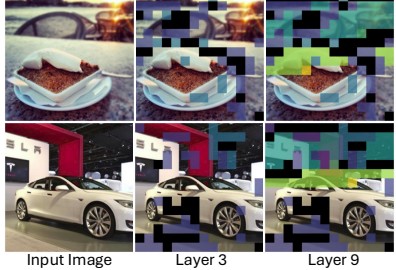

Input Image    Layer 3    Layer 9

Figure 5: Examples of our token condensation. More visualizations can be found in the appendix.

**Visualization of Proposed Token Condensation.** We visualize the pruned and merged tokens of different ViT layers in Figure 5. Here, the black mask indicates pruned regions while different colors are set for different merging clusters. We observe that as token condensation progresses, non-discriminative tokens are gradually removed, leading to better alignment with the text semantics. See Section A.3 for more details.

## 5 CONCLUSION

In this paper, we introduced Token Condensation as Adaptation (TCA), a novel training-free test-time adaptation method for CLIP models. Our comprehensive experiments demonstrated that token condensation significantly benefits the visual-text alignment in CLIP, which can further serve as an interpretation of visual semantics. Additionally, our method reduces GFLOPs as a beneficial byproduct, enhancing computational efficiency. For fair comparisons, we fixed the GFLOPs budget by pre-setting $R$ in our experiments; however, the condensing rate can be adaptively estimated using the distances to merging center as an indicator, which can be a promising direction for future research. We also acknowledge the limitations of TCA as a training-free method, particularly its bottleneck in handling datasets with severe distribution shifts, as discussed in Section A.4.

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

# A APPENDIX

This appendix provides additional descriptions of the proposed TDA, including empirical results and algorithm. Visual aids for token condensation are also included to enhance understanding of the proposed method.

- **Section A.1:** Additional Ablation Study.

- **Section A.2:** Token Condensation Algorithm.

- **Section A.3:** Quantitative study for token condensation ($R = 0.7$).

- **Section A.4:** Potential limitation of TCA.

## A.1 ADDITIONAL ABLATION STUDY

**Impact of Reservoir Size $M$.** We assess the effectiveness of TCA across various reservoir sizes $M$ on Pets, Flower102, and EuroSAT datasets, as illustrated in Figure 4a. Remarkably, although the best performances are achieved at different reservoir sizes for different datasets, our TCA consistently maintains stable and high performance across a wide range of $M$ values. This showcases the robustness and flexibility of TCA with respect to different reservoir budgets. Notably, even under extreme conditions with a minimal reservoir size (*i.e., $M = 1$*), our strategy significantly surpasses the strongest baseline method, TDA, by a large proportion on the EuroSAT dataset (14.2%).

**Impact of Merging Center Number $K$.** We evaluate TCA performance by giving different numbers of merging centers $K$ for Pets, EuroSAT, and Food101 datasets. As shown in Table 7, setting $K = 2$ consistently yields the best results. This choice balances preserving important information and reducing redundancy. A smaller $K$ (*i.e., $K = 1$*) may over-simplify the merging process, leading to the loss of critical

Table 7: Impact of K.

| K | 1 | 2 | 3 | 4 |
|---|---|---|---|---|
| Pets | 89.29 | **89.53** | 89.29 | 89.21 |
| EuroSAT | 66.25 | **70.43** | 66.96 | 67.44 |
| Food101 | 85.15 | **85.31** | 85.31 | 85.38 |

details, especially in diverse datasets like EuroSAT. Conversely, increasing $K$ beyond 2 introduces unnecessary complexity and can over-segment the token space, retaining redundant tokens that contribute little to classification. Therefore, maintaining a very small $K$ (where $K \ll N$) is sufficient and advantageous.

**Impact of Pruning & Merging Ratio.** We experiment with different token pruning and merging ratios under the same computational budget, as shown in Table 8. Incorporating token diversity through merging consistently enhances performance. Specifically, the 2:1 merging-to-pruning ratio outperforms other configurations, especially those favoring pruning. This is because merging preserves diverse token representations by K coresets that pure pruning might dis-

Table 8: Impact of token merging/pruning ratio.

| Merging:Pruning | 0:1 | 1:2 | 2:1 |
|---|---|---|---|
| Pets | 89.04 | 88.99 | **89.53** |
| EuroSAT | 69.63 | 69.98 | **70.43** |

card. When comparing pruning-only (0:1) with the 1:2 merging-pruning ratio on Pets, pruning-only performs better. This may be because the dataset features images with a single prominent object, meaning that pruning background tokens has minimal impact since essential object information remains intact. In contrast, for the EuroSAT dataset, which comprises diverse satellite imagery, simply pruning tokens leads to the loss of important contextual features necessary for accurate classification.

**Impact of Visual Backbone.** Trends similar to ViT-B/16 are observed with the **ViT-L/14** architecture, as shown in Table 9. TCA consistently surpasses TDA across multiple datasets, including Aircraft, Caltech101, EuroSAT, Flower102, Pets, and UCF101, while adhering to a limited GFLOPs budget (19.6% GFLOPs reduction). Even with a 48.9% reduction in GFLOPs, TCA continues delivering satisfactory results. This demonstrates the scalability and robustness of our method across different model sizes, reinforcing its effectiveness without additional training.

**Comparison with State-of-the-Art TTA Using Token Condensation.** We additionally evaluate the performance when combining TDA with token pruning and merging baselines and show the performance gain over TDA + ATS in Figure 4c. Although TDA achieves considerable performance gain, it heavily relies on the negative cache and a large set of hyperparameters. In contrast, TCA's accuracy gain significantly surpasses that of TDA + EViT and TDA + ToME across multiple datasets

Table 9: Results on the cross-dataset benchmark with CLIP ViT-L/14. $^*$ denotes the averaged GFLOPs across all datasets.

| Method | Aircraft | Caltech101 | Cars | DTD | EuroSAT | Flower102 | Food101 | Pets | SUN397 | UCF101 | Average | GFLOPs |
|---|---|---|---|---|---|---|---|---|---|---|---|---|
| CLIP | 31.59 | 94.56 | 78.12 | 57.03 | 63.00 | 79.58 | 90.92 | 93.46 | 69.05 | 76.13 | 73.34 | 81.14 |
| Tent | 27.45 | 94.97 | 76.93 | 57.15 | 66.20 | 74.83 | 89.20 | 93.27 | 68.73 | 75.73 | 72.45 | 81.14 |
| SAR | 26.07 | 94.52 | 75.58 | 56.91 | 63.77 | 75.03 | 89.13 | 93.05 | 68.39 | 75.50 | 71.80 | 81.14 |
| TPT | 30.06 | 95.21 | 76.84 | 52.30 | 55.11 | 76.21 | 88.56 | 93.08 | 67.69 | 73.78 | 70.88 | 143.31 |
| TDA | 33.42 | 95.46 | 78.72 | 57.39 | 66.27 | 79.94 | 90.83 | 93.27 | 70.74 | 78.14 | 74.42 | 81.14 |
| EViT$_{R=0.9}$ | 31.23 | 94.56 | 76.59 | 56.38 | 63.04 | 79.13 | 90.08 | 93.32 | 68.54 | 76.40 | 72.93 | 65.19 |
| ToME$_{R=0.9}$ | 28.29 | 92.54 | 71.26 | 56.68 | 60.30 | 77.87 | 89.77 | 91.28 | 68.21 | 72.22 | 70.84 | 64.74 |
| ATS$_{R=0.9}$ | 25.74 | 93.39 | 67.69 | 55.02 | 52.81 | 76.78 | 86.48 | 91.50 | 66.26 | 72.56 | 68.82 | 43.62$^*$ |
| EViT$_{R=0.7}$ | 26.94 | 92.94 | 62.55 | 53.96 | 52.04 | 73.24 | 80.69 | 90.00 | 63.70 | 71.21 | 66.73 | 40.78 |
| ToME$_{R=0.7}$ | 15.60 | 83.73 | 38.43 | 49.82 | 44.51 | 59.36 | 72.65 | 77.73 | 58.32 | 50.99 | 55.11 | 40.05 |
| ATS$_{R=0.7}$ | 6.87 | 67.87 | 16.37 | 40.78 | 30.12 | 37.43 | 34.50 | 60.94 | 30.07 | 33.44 | 35.84 | 26.76$^*$ |
| TCA$_{R=0.9}$ | 33.84 | 96.39 | 76.93 | 56.38 | 67.74 | 80.71 | 90.21 | 93.54 | 70.02 | 78.24 | 74.40 | **65.24**$_{-19.6\%}$ |
| TCA$_{R=0.7}$ | 29.73 | 94.81 | 63.72 | 53.72 | 60.69 | 76.00 | 81.55 | 90.02 | 65.61 | 73.14 | 68.90 | **41.44**$_{-48.9\%}$ |

---

**Algorithm 1** Token Condensation at the $l$-Layer in $E_v$

---

**Input:**
1: Token reservoir $\mathfrak{R}$;
2: Visual patches $\mathbf{V}^{l-1}$ at layer $l-1$;
3: Pruning threshold $\theta_{\text{prune}}(\alpha \cdot R)$;
4: Merging threshold $\theta_{\text{merge}}(R)$

**Output:** Token-efficient visual feature $\hat{\mathbf{V}}^l$
5: **Token Anchoring**: Obtain $\mathbf{A}_{c^*}^{l-1}$ by Equation (8), using anchor tokens in $\mathfrak{R}$ and sample's `<cls>` token $\mathbf{v}_{\text{cls}}^l$
6: Compute cross-head scores $\mathbf{S}_i^{\text{head}}$ for every token $i$
7: **if** $\forall i, S_i^{\text{head}} \leq \theta_{\text{prune}}(\alpha \cdot R)$ **then**
8:     **Token Pruning**: Obtain $\hat{\mathbf{V}}_{\text{prune}}^l$ via Equation (9)
9: **end if**
10: **if** $\forall i, \theta_{\text{merge}}(R) \leq S_i^{\text{head}} \leq \theta_{\text{prune}}(\alpha \cdot R)$ **then**
11:     **Token Merging**: Obtain $\hat{\mathbf{V}}_{\text{merged}}^l$ via Equation (12)
12: **end if**
13: **return** $\hat{\mathbf{V}}^l$, which is composed of $\mathbf{v}_{\text{cls}}^l$, $\hat{\mathbf{V}}_{\text{prune}}^l$ (excluding merged tokens), and $\hat{\mathbf{V}}_{\text{merged}}^l$

---

and on average, even with a minimal set of hyperparameters, highlighting its superior adaptation capability.

## A.2 ALGORITHM

**Algorithm 1** outlines a simple process for performing token pruning and merging at layer $l$ in a ViT-based CLIP model. We first obtain the averaged anchor token $\mathbf{A}_{c^*}^{l-1}$ by the `<cls>` tokens saved in the reservoir $\mathfrak{R}$. Token condensation is then conducted given the anchor token. Specifically, we conduct token pruning by relative ranking positions of token $i$ across multiple attention heads. Then, coreset selection is used for token merging. Finally, we concatenate the `<cls>` token $\mathbf{v}_{\text{cls}}^l$ with the retained tokens as the input for the next layer, where the original $N + 1$ tokens are shrunk to $(R \cdot N) + 1$, thereby reducing the computational cost.

## A.3 QUANTITATIVE STUDY

We visualize the token condensation masks at layer 3, layer 6, and layer 9, and compare them with the original image across multiple datasets, as shown in Figure 6. As the layers go deeper, we observe that class-irrelevant patches are gradually pruned, as indicated by the black mask. TCA also merges class-ambiguous patches, such as fur in cat images, and ground and sky in aircraft and car images. All similar tokens are merged into a single token using our proposed coreset selection strategy. After token condensation, the sample features retain only discriminative information, thereby bridging the gap between visual and text features, and mitigating the distribution shift between pretrained data and unseen datasets.

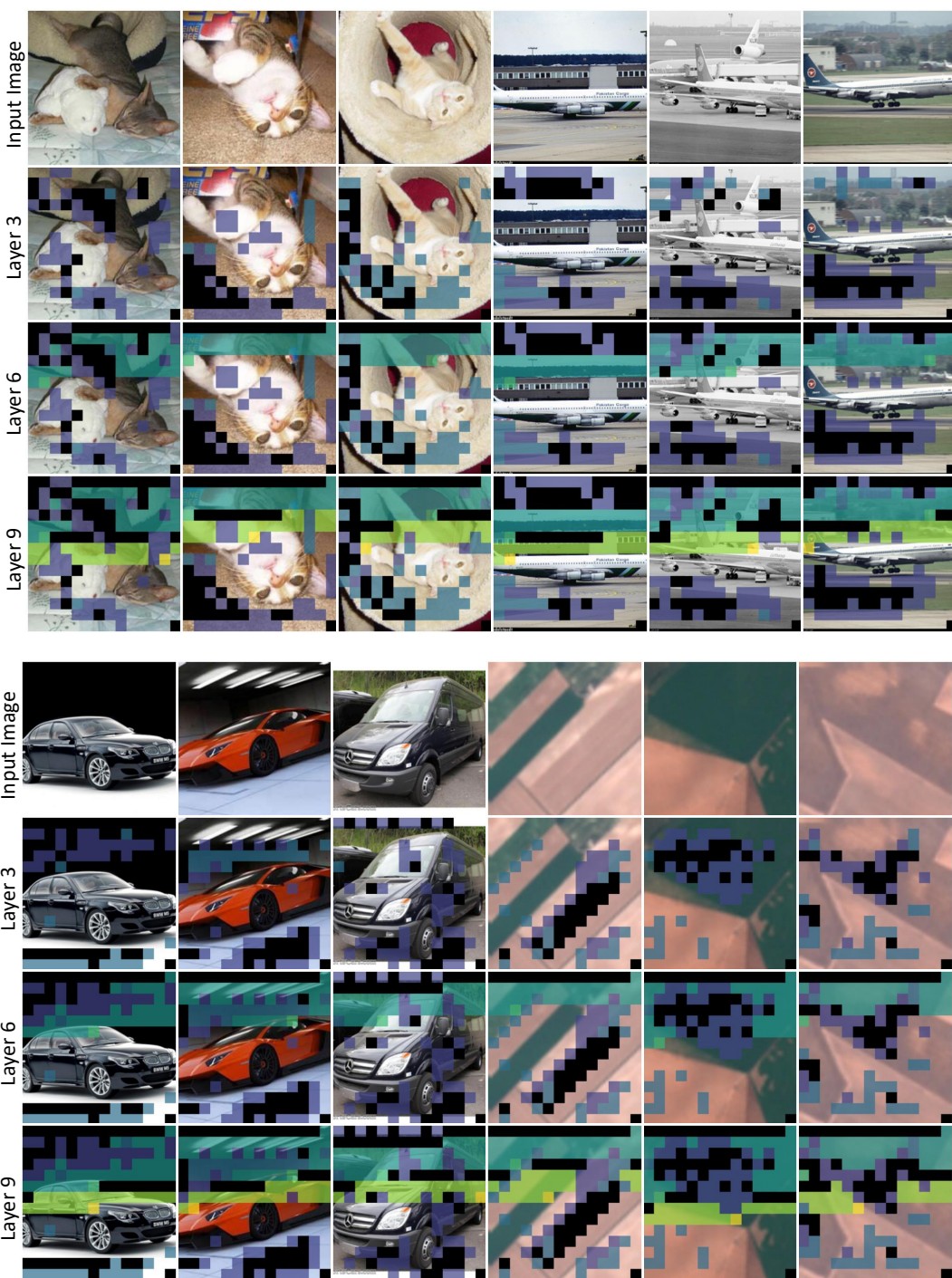

Figure 6: Visualization of our proposed token condensation with $R = 0.7$. Pruned tokens are masked in black, while different colors represent distinct merging clusters.

### A.4 DISCUSSION ON THE LIMITATION OF TCA

In this section, we discuss the potential limitations of our proposed TCA. Due to the training-free nature of the approach, it is challenging to mitigate the performance gap when the testing domain diverges significantly from the training domain. As observed in the out-of-distribution (OOD) samples shown in Figure 7, the ground truth object is not always centrally located, and larger class-irrelevant

Figure 7: **Sample data from the OOD benchmark.** The samples from the same class exhibit significant diversity. For instance, in the ImageNet-R dataset, one image of a great white shark is dominated by shoes and human legs, while another is on top of a building, showing extreme variability.

objects (*e.g.,* humans or shoes) can sometimes dominate the prediction. This issue is particularly prominent in CLIP models, where text features for all classes are predefined. When the dominant object is included in the label set, accurately directing visual features to the correct class without additional training becomes difficult. Moreover, the diversity of OOD samples introduces further complexity, especially in the absence of data augmentation. These observations raise important questions for future research: (1) How can we quantify the capacity to mitigate domain shift effectively? (2) What lightweight solutions can be developed for backpropagation and network updates to facilitate test-time adaptation? We leave these questions for future work.

