# OpenReview forum: "Tokens on Demand: Token Condensation as Training-free Test-time Adaptation"
_ICLR.cc/2025/Conference — ICLR 2025 Conference Withdrawn Submission_

### Official Review · Reviewer_cmcN · 2024-10-21

**Soundness:** 2
**Presentation:** 1
**Contribution:** 2
**Rating:** 3
**Confidence:** 3

**Summary:**

This post introduces a new method called Token Condensation as Adaptation (TCA) that aims to address the distribution shift problem encountered by Visual Language Models (VLMs) during test-time reasoning. TCA addresses the problem of visual distribution shift by compressing image tokens that have little attention from the <cls> token. TCA proposes a contextual token reserve (CTR) to identify and track the most reliable <cls> tokens in the historical data stream that are aligned with the target category. CTR retains tokens with the lowest uncertainty as "anchors" to guide the retention of category-related tokens during reasoning. At the same time, these anchors in turn act as token-level classifiers to correct the prediction results of the VLM.

**Strengths:**

1. The proposed method can reduce the computation overhead of ViT.

**Weaknesses:**

1. The structure of the article needs further optimization.
2. Some concepts lack necessary explanation.

**Questions:**

1. The structure of the article needs further optimization:

    a) In lines 71-72, there is a lack of explanation for the experiment in Fig. 2, and when reading this part, it is difficult to understand how the authors reached their conclusions. From Fig 2(a), we can only see that different tokens have different responses to <cls> token. Similarly, without explaining clearly how the anchor token is obtained and used, the author explains the role of the anchor in lines 83-84. These make reading difficult.

    b) The summary of the method proposed in the paper in the introduction is too abstract, making it difficult for readers to grasp the actual method proposed by the author.

    c) The best resutls in Tab.1 and 2 should be highlighted.

2. The concept of visual shift needs further clarification. What is the connection between the proposed method and domain shift? Why can ``visual shift'' be reduced by removing task-irrelevant features? In my opinion, the method proposed by the author only eliminates interference by removing task-irrelevant features.

3. The method proposed by the authors is related to the order of the input, and related ablation experiments are required, such as the impact of the shuffle test set on performance.

---

### Official Review · Reviewer_8qmQ · 2024-10-31

**Soundness:** 3
**Presentation:** 4
**Contribution:** 3
**Rating:** 5
**Confidence:** 4

**Summary:**

The authors proposed Token Condensation as Adaptation, a token pruning and merging method that operates at test-time on the ViT blocks in VLMs. Token condensation was previously framed as a way to reduce computational burden for vision transformers by either removing unimportant tokens or merging similar tokens. The authors, on the other hand, proposed to use token condensation as a measure to address multimodal distribution shifts by attenuating the effect of irrelevant and ambiguous tokens.
The authors showed through experiments that their method both reduces GFLOPs and improves performance. The experiments include results on a wide range of datasets. The baselines are rather exhaustive. Results are somewhat promising.

**Strengths:**

1. The idea of using token condensation for improving distribution shift is novel and encouraging. The authors provided good intuition to how their method works. The visualizations of the three types of tokens are generally convincing.
2. Experiments cover a wide range of datasets and settings, although the strength of the method was not reflected in all of them.

**Weaknesses:**

1.   Most of the discussions are centered around CLIP. The reviewer believe the paper would be more impactful if the authors could share thoughts on extending their method to other settings, such as integration with recent open-source large VLMs. Despite the novelty presented in the paper, test-time adaptation on CLIP is already a rather extensively-studied area. If CLIP is the only experimental setting, it would be questionable if TCA could bring significant impact to the VLM research community.
     1.   For the reason above, the paper’s abstract and introduction sections contain rather too ambitious writings (compared to the conclusion section where the authors downgraded their method to yet another “test-time adaptation method for CLIP models.) These sections need a major revision.
2.   Experimental results are weak in the OOD setting. The authors provided brief explanation for this in the appendix; however, I believe this would not be sufficient because the proposed method was supposed to counter distribution shift. The paper needs to address the problem of how their method, possibly while work together with another method, could bring significant improvement in the OOD setting. If we think about a industry user or future researcher of CLIP, is such performance improvement sufficient evidence to justify using the TCA method? Considering that TCA requires a major change of the code and may not necessarily accelerate computation.
3.   Results in table 1 show high unevenness across datasets. Any possible causes?

The reviewer is somewhat satisfied with the results already presented in the paper, but have concerns on its application and impact. I look forward to the authors’ response.

**Questions:**

I would like to learn more about how the implications of those GFLOPs reductions from the proposed method. For example, how does it affect the latency of the model? How does the Context Token Reservoir affect the latency?

---

### Official Review · Reviewer_vH1N · 2024-11-02

**Soundness:** 3
**Presentation:** 2
**Contribution:** 2
**Rating:** 5
**Confidence:** 3

**Summary:**

The author introduces a new method named Token Condensation as Adaptation (TCA) to address distribution shift issues faced by Visual Language Models (VLMs) during test-time reasoning. TCA tackles the visual distribution shift problem by compressing image tokens that receive minimal attention from the <cls> token. The author proposes a contextual token reserve to identify and track the most reliable <cls> tokens from historical data streams aligned with the target category. CTR retains tokens with the lowest uncertainty guiding the retention of category-related tokens during reasoning.

**Strengths:**

1. The paper is well written and easy to follow.

**Weaknesses:**

1. The novelty is limited. Using the [cls] token to select tokens is not a novel approach, as it has already been validated in many vision methods[1]. Therefore, I cannot recognize it as an innovation.

2. In Table3, The TCA surpasses ToMe by a very little margin. This does not sufficiently highlight the advantages of TCA.

3. The scope of tasks is too limited. The author only validates TCA on classification tasks, while VLMs encompass many other tasks. Rather than being a broad enhancement for VLMs, the method seems more like an improvement specific to CLIP's vision encoder. Could the author validate the effectiveness of the proposed method on a broader range of tasks?

[1]EViT: Expediting Vision Transformers via Token Reorganizations, ICLR2022

**Questions:**

Please refer to the weakness.

---

### Official Review · Reviewer_stTV · 2024-11-03

**Soundness:** 2
**Presentation:** 3
**Contribution:** 3
**Rating:** 5
**Confidence:** 4

**Summary:**

This paper presents Token Condensation as Adaptation (TCA), a training-free method designed to handle distribution shifts in vision-language models (VLMs) during test-time inference. TCA condenses image tokens by focusing on those with significant attentiveness to the <cls> token. The method uses a context token reservoir (CTR) to store tokens with low uncertainty as anchors, which help retain class-relevant information and guide inference. The results show that TCA enhances cross-dataset and out-of-distribution adaptation performance, reducing computational requirements.

**Strengths:**

* The flow of the argument is solid and novel.
     * Specifically, the finding that not all visual patches are useful for visual representation and taking this motivation to exclude these from the test-time adaptation makes sense and is interesting.
     * Also, finding a better <cls> token that only attends to its target class instead of other concepts is an exciting idea grounded in logical reasoning.

* The biggest hurdle of previous Test-time adaptation methods for CLIP is their heavy computational cost (augmentation and backprop time). Although the proposed method does not necessarily beat the performance of previous Test-time methods for CLIP, the proposed method is augmentation-free and thus significantly reduces the computational cost needed, which is valuable for the test-time use case.

* The paper is formatted well with good figures.

**Weaknesses:**

* In Figure 2-(a) x-axis, state the ordering of the sorting (e.g., highest to lowest)

* State what the different colors mean in the figure with legends or in the figure caption.

* In the Ablation study, the authors showed the impact of the different $\beta$ and $\lambda$ terms on their performance. What values were used for the main experiments in Table 1 and Table 2? Also, since these are hyperparameters, wouldn't the authors' claim in line 375 that the proposed method is hyperparameter-free be wrong?

* The conclusions drawn in this work are empirical rather than theoretical. While basing a method on empirical observations is valid, the absence of theories  somewhat decreases the overall contribution of the research.

**Questions:**

See weaknesses above.

---

### Note · Authors · 2024-11-13

I have read and agree with the venue's withdrawal policy on behalf of myself and my co-authors.